# MODEL-BASED SALIENCY FOR THE DETECTION OF ADVERSARIAL EXAMPLES

## ABSTRACT

Adversarial perturbations cause a shift in the salient features of an image, which often results in misclassification. Previous work has suggested that these salient features could be used as a defense, arguing that with saliency tools we could successfully detect adversarial examples. While the idea itself is appealing, we show that prior work which used gradient-based saliency tools is ineffective as an adversarial defense – it fails to beat a simple baseline which uses the same model but with the saliency map removed. To remedy this, we demonstrate that learnt saliency *models* can capture the shifts in saliency due to adversarial perturbations, while also having a low computational cost. This allows saliency models to be used effectively as a real-time defense. Further, using the learnt saliency model, we propose a novel defense: a CNN that distinguishes between adversarial images and natural images using salient *pixels* as its input. On MNIST, CIFAR-10, and ASSIRA, our defense improves on using the saliency map alone, and can detect various adversarial attacks. Lastly, we show that even when trained on weak defenses, we can detect adversarial images generated by strong attacks such as C&W and DeepFool.

## 1 INTRODUCTION

Adversarial examples highlight a crucial difference between human vision and computer image processing. Often computers fail to understand the relevant characteristics of an image for classification (Ribeiro et al., 2016) or fail to generalize locally, i.e., misclassify examples close to the training data (Szegedy et al., 2013). Attacks exploit this property by altering pixels the classifier heavily relies on – pixels which are irrelevant to humans for object recognition. As a consequence, adversarial perturbations fool classifiers while the correct class remains clear to humans.

Saliency maps identify the pixels an image classifier uses for its prediction; as such, they can be used as a tool to understand why a classifier is fooled. Building on this concept, researchers have shown qualitatively that adversarial perturbations cause a shift in the saliency of classifiers (Fong & Vedaldi, 2017; Gu & Tresp, 2019). Figure 1 shows examples of a natural image and corresponding adversarial images, each above their respective saliency maps. The saliency maps corresponding to adversarial images show perceptible differences to that of the original image, even though adversarial images themselves often seem unperturbed. For the original image, the saliency map shows that the classifier focuses on the four (and a couple of random pixels on the left). We observe that for the adversarial images, the classifier starts focusing more on irrelevant aspects of the left side of the image.

There is ample research into different techniques for finding saliency maps (see e.g. Zeiler & Fergus, 2014; Springenberg et al., 2014; Bach et al., 2015; Ribeiro et al., 2016; Shrikumar et al., 2017; Selvaraju et al., 2017; Zintgraf et al., 2017; Fong & Vedaldi, 2017). However, not all saliency maps are equally informative (Fong & Vedaldi, 2017). For example, the Jacobian[1] can be used to determine the saliency of a pixel in the classification of the image (Papernot et al., 2016b; Zhang et al., 2018). As the Jacobian is often used to generate adversarial examples, intuitively, we expect that it can be used effectively to detect adversarial perturbations. Zhang et al. (2018) propose a defense to this effect: they determine whether an input is adversarial, given the Jacobian-based

---

[1]i.e. the forward derivative of the classifier with respect to the input image

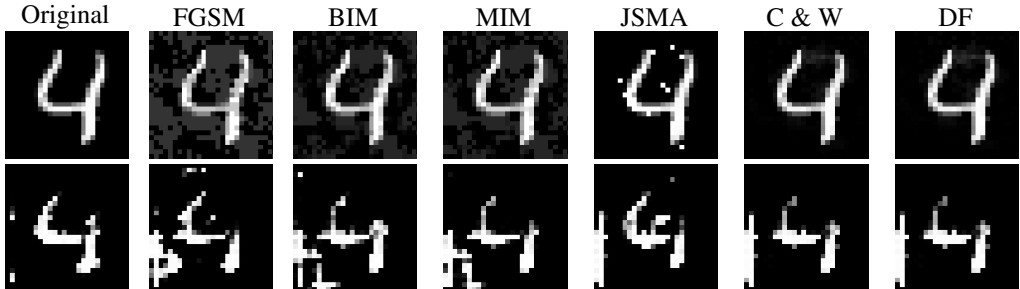

Figure 1: Example of the saliency maps of natural and adversarial images for the MNIST dataset. The top is the input image and the bottom shows the corresponding saliency map. In the second row, lighter colours correspond to higher saliency (black corresponds to a saliency of 0, the lowest possible value). The classifier predicts (from left to right) the images as: 4, 9, 9 , 8, 9, 9. Note the stark difference between the saliency masks of the original image and those of the adversarial examples.

saliency map concatenated *with the image*. However, as shown *qualitatively* by Gu & Tresp (2019), gradients are not always able to capture differences between adversarial images and natural images (for an example see Figures 7 and 8 in Appendix D).[2] Here we inspect the proposed Jacobian-based approach and show that only the concatenated input affects the technique's performance in detecting adversarial examples, with the Jacobian having no effect.

While gradients may not be informative for detection, saliency should be an effective tool for detecting adversarial images. In our analysis, we use more powerful model-based saliency techniques and show that the magnitude of the shift of the saliency map due to adversarial perturbations often exceeds the $L_2$ distance between the saliency maps of different natural images. Building on this result, we consider two different possible effects adversarial perturbations might have on the classifier:

1. They might cause the classifier to focus on the *wrong pixel locations*
2. They might *change the pixel values* of salient pixels

Based on these hypotheses, we employ two CNN classifier architectures to detect adversarial images. Claim (1) can be captured by shifts in saliency maps, as previously considered by Fong & Vedaldi (2017). In this work, we extend on their analysis[3] by proving the defensive capability of our model-based saliency against difficult black-box attacks, such as C&W and DeepFool[4], as well as white-box adversarial attacks. By considering claim (2), we demonstrate that incorporating pixel values improves the performance of the classifier when shifts in saliency maps do not suffice to capture adversarial perturbations. We also show that our salient-pixel based defense generalizes well (detecting stronger attacks when trained on weaker attacks) and is more robust than the saliency map defense against white-box attacks. Lastly, we demonstrate that saliency can be used to detect adversarial examples generated by small perturbations, contrary to other defenses, which exhibit *threshold behavior*: i.e., when the adversarial perturbation is too small, other defenses (specifically Gong et al., 2017; Zhang et al., 2018) are unable to detect the adversarial images.

## 2 SALIENCY AND ADVERSARIAL EXAMPLES

Saliency maps and adversarial perturbations have similar mathematical formulations and derivations. Both are computed by investigating the relation between the values of pixels and the classification score. Adversarial examples are found by deriving the minimal perturbations required to change the classification of an image. Saliency is computed by finding the pixels used by the model

---

[2]Similarly, Fong & Vedaldi (2017) show that gradient-based heat maps are less effective than other saliency methods in detecting adversarial perturbations generated using BIM (Kurakin et al., 2016).

[3]Their main contribution is that saliency maps generated by different techniques are not equally effective in capturing changes due to adversarial perturbations (produced using BIM (Kurakin et al., 2016).

[4]These attacks generate smaller $L_2$ perturbations, making them more difficult to detect. The perturbation size used by Fong & Vedaldi (2017) can likely still be detected by a simple classifier that trains on images.

to determine the class of an object (Simonyan et al., 2013). Saliency maps can be found by considering the smallest part of an image that is sufficient for a correct classification, known as the *smallest sufficient region* (SSR), or whose removal is sufficient for an incorrect classification, known as the *smallest destroying region* (SDR) (Dabkowski & Gal, 2017; Fong & Vedaldi, 2017). Observe that the latter definition of saliency is very close to that of adversarial examples.

Mathematically, both saliency maps and adversarial perturbations can be derived in a similar fashion. Consider adversarial examples. The general formulation of an adversarial attack can be summarized as follows:

$$\min_r ||r||_k$$
$$\text{s.t. } f(x + r) = y' \neq y \qquad (1)$$
$$x + r \in [0, 1]^m$$

where $x$ is the natural image, $r$ is the adversarial perturbation, $y$ is the correct class, and $y'$ is an incorrect class. Due to the non-linearity of NNs, solving the above problem requires non-linear optimization. Therefore, in practice several different approaches to solving the above formulation have been implemented. For example, Goodfellow et al. (2014) set $r = \varepsilon \text{sign}(\frac{\delta f(x)}{\delta x})$. Similarly, saliency can be computed using the forward derivative $\frac{\delta f(x)}{\delta x}$ (Papernot et al., 2016b; Zhang et al., 2018).

Previous research has already started investigating the relation between saliency and adversarial examples. This includes:

**Using saliency to attack** Researchers have devised adversarial attacks that use saliency (Papernot et al., 2016b; Yu et al., 2018). The key idea is to use saliency to determine the pixel that is most sensitive to perturbation iteratively. The main benefit is that fewer pixels are perturbed – often perturbing as few as $4\%$ of the pixels suffices to change the classification of the image (Papernot et al., 2016b).

**Using saliency to defend** Fong & Vedaldi (2017) introduce a method that detects adversarial perturbations by using heat-map visualizations of the predicted class. However, in their analysis, they only use BIM (Kurakin et al., 2016), which is easily detected. Further, Zhang et al. (2018) hypothesize that there is a mismatch between the saliency of a classification model and the adversarial example. They propose a defense against adversarial attacks by training a classifier on images concatenated with their saliency map, which is computed by calculating the Jacobian of the classifier with respect to the image $x$, i.e., $s_x = \nabla_x f(x)$. Zhang et al. (2018) find that their method obtains a high accuracy (often near $100\%$) when detecting adversarial images generated by FGSM, MIM, and C&W attacks on MNIST, CIFAR-10, and 10-ImageNet. However, Gu & Tresp (2019) contradict these results, and demonstrate that the gradients show imperceptible differences due to adversarial perturbations (see Figures 7 and 8 in Appendix D).

**Adversarial robustness and interpretability of models** Fong & Vedaldi (2017) and Gu & Tresp (2019)[5] show that saliency maps can be used to explain adversary classifications. Both highlight an important trend: *not all techniques used to compute saliency maps show shifts in saliency maps due to adversarial perturbations*. Further, Tsipras et al. (2018) shows that more robust models have more interpretable saliency masks. Etmann et al. (2019) quantify the relation by investigating the alignment between the saliency map and the input image.

## 3 MODEL-BASED SALIENCY FOR THE DETECTION OF ADVERSARIAL EXAMPLES

In this section, we explain how we construct and evaluate our saliency-based adversarial example detectors.

---

[5]Work done by Gu & Tresp (2019) was in parallel.

### 3.1 CLASSIFIER

We train a convolutional neural network image classifier, which we target with black-box attacks; the architectures are summarized in Appendix A. We use cross-entropy loss and optimize the parameters using Adam (Kingma & Ba, 2014) with the standard hyperparameters settings.

### 3.2 BLACK-BOX ATTACKS

In our analysis, we consider six different adversarial attacks: Fast Gradient Sign Method (FGSM), Basic Iterative Method (BIM), Momentum Iterative Method (MIM), $L_2$ Carlini & Wagner (2017b) (C&W), Jacobian-based Saliency Map Approach (JSMA) and DeepFool (DF) (for each attack see Goodfellow et al., 2014; Kurakin et al., 2016; Dong et al., 2018; Carlini & Wagner, 2017b; Papernot et al., 2016b; Moosavi-Dezfooli et al., 2016, respectively). We use the implementation as provided in **cleverhans** (Papernot et al., 2016a). The hyper-parameters are summarized in Appendix B.

### 3.3 MODEL-BASED SALIENCY

To generate saliency masks, we adapt the method used by Dabkowski & Gal (2017). Our reason is twofold: the technique computes high-quality saliency masks at a low computational cost. Dabkowski & Gal (2017) employ a U-Net with a novel loss function that targets SDR, SSR, mask sparsity, and mask smoothness. We adapt the original loss function to omit the total variational term, as mask smoothness is not required in our analysis.

Let $f_s = f_s(x, f_c(x))$ denote the generated map. First, the map average $AV(f_s)$ is used to ensure that the area of the map is small. Second, $\log(f_c(\Phi(x, f_s)))$ is included to ensure that the salient pixels suffice to identify the correct class. Finally, $f_c(\Phi(x, 1 - f_s))$ is included to ensure that the classifier can no longer recognize the class if the saliency map is removed. Therefore, our saliency loss function is:

$$L(f_s) = \lambda_1 AV(f_s) - \log(f_c(\Phi(x, f_s))) + \lambda_2 f_c(\Phi(x, 1 - f_s))^{\lambda_3}, \tag{2}$$

where $f_c$ is the softmax probability of the class $c$, $\Phi(x, f_s)$ applies mask $f_s$ to image $x$, and $\lambda_i \geq 0$ are hyper-parameters.

We adapt the PyTorch implementation provided by Dabkowski & Gal (2017)[6] and train the saliency model on standard, non-adversarial images only. For evaluation, we use the same saliency model for both natural and adversarial images. When generating the saliency maps for our images, we use the predicted classification for feature selection to prevent an information leak (which would occur if we use the true label).

### 3.4 DEFENSE

Our hypothesis is that if an image is adversarial, the classifier likely focuses on the *wrong aspects* or the *pixels on which it focuses are misleading* (due to the perturbed color or intensity) when classifying an image as adversarial. We consider two different cases by building classifiers for (1) *saliency maps* and (2) *salient pixels*. For both classifiers, we use the same architecture (and hyper-parameters) as for the black-box image classifiers (as summarized in Appendix A).

#### 3.4.1 SALIENCY MAP: "THE CLASSIFIER FOCUSES ON THE WRONG PIXEL LOCATIONS"

We build a detector based on the saliency maps of images as follows. First, we train a classifier and generate adversarial images for every natural image in the training dataset. Then we generate the saliency maps for the clean data $\{f_s(X)\}$ and adversarial images $\{f_s(X_{adv})\}$. We build a binary detector for the saliency maps, which predicts whether the corresponding image is adversarial or natural. We abbreviate this defense as SMD (**S**aliency **M**ap **D**efense). We do not concatenate the saliency maps to the input image.

---

[6]The code is available at https://github.com/PiotrDabkowski/pytorch-saliency.

### 3.4.2 SALIENT PIXEL: "THE PIXEL VALUES ARE WRONG"

We construct a second classifier for the *salient pixels*. We follow the same steps as outlined in the previous section, aside from the final step. We define the salient pixels as $f_s(x) \cdot x$, where $x$ is the image, $f_s(x)$ is the saliency map corresponding to $x$ and $\cdot$ denotes the element-wise product. We abbreviate this defense as SPD (**S**alient **P**ixel **D**efense). Similarly to SMD, we do not concatenate the saliency maps to the input image.

### 3.5 BENCHMARKS AND BASELINES

To benchmark our results, we consider two baselines. First, we train a baseline classifier that classifies input as adversarial or natural based on the images alone. This allows us to evaluate the added benefit of using saliency maps. This method was implemented by Gong et al. (2017). We abbreviate this defense as ID (**I**mage **D**efense).

Second, we compare our defense method with the saliency-based defense of Zhang et al. (2018) (see Section 2). We abbreviate this defense as JSD, for **J**acobian-based **S**aliency map **D**efense. In our implementation, we adapt the method of Zhang et al. (2018); we find that if we use $f_s(x) = \nabla_x f(x)$ as the saliency map it leads to underflow, resulting in a zero matrix. Therefore, instead we take the derivative with respect to the logits, i.e. $f_s(x) = \nabla_x z(x)$.

JSD is mathematically related to the other defenses. First, it is more general compared to ID: the filters of JSD can learn to ignore the Jacobian-based saliency, in which case the two methods are equivalent. Further, JSD is similar to SMD, as the filters can learn to ignore the image input. In this case, the only difference between JSD and SMD is that they use different techniques to derive saliency. However, JSD differs from SPD, as CNN filters cannot multiply one channel by another.

### 3.6 EVALUATION WITH BLACK-BOX ATTACKS

We follow the evaluation protocol of Zhang et al. (2018) and train each defense to detect adversarial images generated by a specific attack, thereby generating six different detection models (one for each black-box attack). To generate the training data, we generate one adversarial example for every clean image. The training data becomes $[X, X_{adv}]$, where $X$ denotes the clean data and $X_{adv}$ denotes the adversarial data, and the labels are $[\mathbf{1}_n, \mathbf{0}_n]$, $\mathbf{1}_n$ and $\mathbf{0}_n$ are one- and zero-vectors of length $n$, respectively. We use the same training procedure and models, as summarized in Appendix A, and report the accuracy of the classifiers on the test dataset.

We compare the performance of the models on MNIST, CIFAR-10 and ASSIRA (see Burges & Cortes, 1998; Krizhevsky et al., 2009; Elson et al., 2007, respectively). In addition to the two frequently used benchmarks, we consider the ASSIRA cats and dogs dataset[7] as it contains high-quality images but is less computationally expensive than ImageNet.[8] Further details on the datasets can be found in Appendix A.

### 3.7 WHITE-BOX ATTACK

Many defenses hold up against black-box attacks but often are unable to defend against white-box attacks (Carlini & Wagner, 2017a). For this reason, we generate white-box attacks tailored to the defense strategy. Our white-box attacks are iterative gradient-based attacks, which target both the classifier and the defense. Inspired by FGSM, we can target the classifier $f$ as

$$x_{adv} = Clip(x + \varepsilon \text{sign}(\nabla f(x_{adv}))) \tag{3}$$

and the defense $d$ as

$$x_{adv} = Clip(x + \varepsilon \text{sign}(\nabla d(x_{adv}))), \tag{4}$$

where $Clip$ clips the pixels to the pre-defined maximum range. Using the above idea, we iterate between Equations 3 and 4 to generate the white-box attack for ID (the defense based on image

---

[7]with the images scaled to $112 \times 112 \times 3$ for computational reasons.

[8]In particular, JSMA would have required substantial time; other researchers have omitted this attack on ImageNet due to the high computational cost that results from the high resolution of the images (Carlini & Wagner, 2017b).

classification). We propose similar white-box attacks for the other defenses, as shown in Appendix C. We limit the number of iterations $T$ to 5, as we find it to be adequate to generate a sufficiently strong attack and further increasing $T$ does not improve the performance.

Our method is similar to that of Metzen et al. (2017). They propose finding adversarial examples as:

$$x_{adv} = Clip(x_{adv} + \varepsilon(\alpha\text{sign}(\nabla f(x_{adv})) + (1 - \alpha)\text{sign}(\nabla d(x_{adv}))), \qquad (5)$$

where in our case $\alpha = 0.5$. The key difference is that we iterate between Equations 3 and 4, rather than applying 3 and 4 simultaneously. We find that this is more effective at targeting the defense, which is more difficult to fool than the original classifier.

## 4 DEFENSE RESULTS

We start by assessing the shift in saliency maps generated by adversarial perturbations and then present the efficacy of the detector against different adversarial attacks. Details, such as attack success rate, can be found in Appendix B.

### 4.1 $L_2$ DISTANCES BETWEEN SALIENCY MAPS

We start by quantifying the shift in saliency maps due to adversarial perturbations; we compute the $L_2$ distance between saliency maps of a natural image and its corresponding adversarial image. As a baseline, we compare these values with the $L_2$ distance between two different natural images. These statistics are summarized in Table 5.

For CIFAR-10 and ASSIRA, the $L_2$-norm between the saliency maps of a natural image and its corresponding adversarial image is comparable to or larger than the $L_2$ distance between two different natural images. Using a Mann-Whitney U-test, we prove quantitatively that the shift is significant for most adversarial attacks on CIFAR-10 and ASSIRA images. This suggests that our saliency-based method is an effective way of capturing adversarial perturbations.

Table 1: $L_2$ distance between (1) saliency maps of different images (row labelled *Different Images*) and (2) the saliency maps of natural images and the adversarial image (generated by the type of attack specified in the row). **The entries correspond to MNIST/CIFAR-10/ASSIRA**. The p-value is derived using the Mann Whitney U-test, where we test whether the sample of $L_2$ distances between a natural and adversarial image is from the same distribution as different images. We use a non-parametric test to avoid assuming normality of the data.

| | STATISTIC | | |
| --- | --- | --- | --- |
| | MEAN $(10^0/10^{-1}/10^{-1})$ | STD $(10^0/10^{-1}/10^{-1})$ | P-VALUE |
| DIFFERENT IMAGES | 0.13/0.75/0.26 | 0.03/0.28/0.03 | |
| FGSM | 0.09/0.83/0.29 | 0.03/0.27/0.03 | 0.00/0.00/0.00 |
| BIM | 0.08/0.76/0.29 | 0.03/0.27/0.03 | 0.00/0.31/0.00 |
| MIM | 0.08/0.76/0.29 | 0.04/0.27/0.03 | 0.00/0.39/0.00 |
| C & W | 0.11/0.80/0.29 | 0.04/0.27/0.03 | 0.00/0.00/0.00 |
| DF | 0.07/0.81/0.30 | 0.03/0.27/0.03 | 0.00/0.00/0.00 |
| JSMA | 0.10/0.88/0.29 | 0.03/0.27/0.03 | 0.00/0.00/0.00 |

### 4.2 BLACK-BOX DEFENSE

Figure 2 summarizes the performance of the defense models trained on a single adversarial attack on different adversarial attacks; the values and standard deviations can found in Appendix G. The overall performance of the model-based saliency defense suggests that saliency can be used to determine whether an image is adversarial.

**Salient Pixel Defense outperforms Saliency Map Defense** Overall, SPD (shown in blue) outperforms the other defenses, suggesting that the salient pixels provide useful information to the detector. Further, our defense generalizes well: even when trained on a weaker attack, SPD is able to detect stronger attacks. Both baseline methods, ID and JSD, only generalize well when trained on a stronger attack. When trained on a weaker attack, they are not able to detect stronger adversarial attacks.

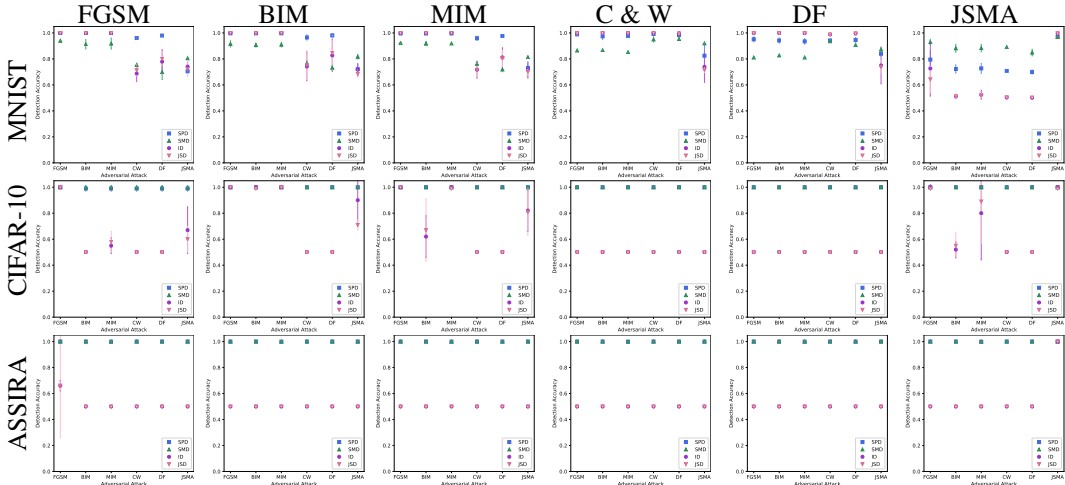

Figure 2: From left to right: performance of SPD, SMD, ID and JSD in detecting adversarial examples when trained on FGSM, BIM, MIM, DF, CW and JSMA. Rows correspond to MNIST, CIFAR-10 and ASSIRA, respectively.

**Jacobian-based Saliency is Uninformative**    The method introduced by Zhang et al. (2018) does not perform well – the (Jacobian) saliency mask defense does not improve on an image-based defense.[9] A congruent observation was made *qualitatively* by Gu & Tresp (2019), who show that the changes in gradient due to adversarial perturbations are imperceptible (for an example, see Figures 7 and 8 in Appendix D). Our quantitative results build on their observation, showing that not all saliency methods show shifts due to adversarial perturbations.

**Worse generalization on JSMA**    We observe a drop in performance of the models when detecting JSMA, likely because JSMA is an $L_0$-norm attack, which generates a different type of adversarial examples. This may suggest that defenses trained on a specific norm, only generalize well to other attacks generated by a norm that produces similar perturbations. FGSM, BIM, and MIM are $L_\infty$-norm attacks, and C &W and DF are $L_2$-norm attacks. Both generate perturbations that are spread out over the entire image, contrary to $L_0$ norm attacks, which changes a few pixels using larger perturbations.

**Threshold Behavior**    Both ID and JSD exhibit *threshold behavior*: they are unable to detect adversarial examples if the perturbation size is below a given threshold. For example, see the performance of both defenses on the ASSIRA dataset. There is a strong correlation between detection accuracy and perturbation size, as measured by $L_\infty$ and $L_2$ (see Table 2). ID is able to detect all adversarial images for which the perturbation size is either $L_2 > 0.027$ or $L_\infty > 0.50$, such as FGSM and JSMA.[10] However, the perturbations are much smaller for DF and CW, making these attacks harder to detect. The threshold appears to occur around $L_2 = 0.025$, as ID can sometimes detect the FGSM perturbations, generated with this size. This observation is in line with the results of Gong et al. (2017), who find that ID is highly efficient at detecting adversarial images with perturbations of $\varepsilon \geq 0.03$ but unable to detect adversarial perturbations generated using $\varepsilon = 0.01$ (using FGSM for images scaled between 0 and 1), obtaining an accuracy of $50.0\%$ in the latter case.[11]

---

[9]We observe that JSD performs similarly, although sometimes worse, compared to ID. Theoretically, the parameter space of ID is a subset of the parameters of JSD. The additional input (the Jacobian) makes the model more difficult to train. Therefore, the difference in results can be attributed to training: the model is more difficult to train due to the increased number of parameters and does not learn to ignore the additional input.

[10]FGSM is known to generate large perturbations. The perturbations for JSMA are relatively large as the attack minimizes the $L_0$ norm, thereby perturbing as few pixels as possible, but by a large amount.

[11]Our perturbations for FGSM are larger than 0.01 to ensure that FGSM is sufficiently strong (see Appendix B for a summary of the attack success rates).

Table 2: Perturbation Sizes of Different Attacks. **Entries correspond to MNIST/CIFAR-10/ASSIRA**. Entries for which $||r||_2 \geq 0.025$ and $||r||_\infty \geq 0.5$ are in bold, where $r$ is the adversarial perturbation. $m$ is the number of pixels in an image.

| ATTACK | PERTURBATION RATE | PERTURBATION SIZE | |
|---|---|---|---|
| | $(L_0/m)$ | $L_2$ | $L_\infty$ |
| FGSM | 0.61/0.94/0.99 | **0.15**/**0.19**/**0.025** | 0.20/0.20/0.025 |
| BIM | 0.64/0.82/0.81 | **0.09**/**0.06**/0.013 | 0.15/0.15/0.025 |
| MIM | 0.62/0.91/0.93 | **0.10**/**0.11**/0.019 | 0.15/0.15/0.025 |
| C & W | 0.99/1.00/1.00 | **0.027**/0.0017/0.0017 | 0.32/0.013/0.037 |
| DF | 0.61/0.83/0.99 | **0.062**/0.0019/0.0020 | 0.45/0.013/0.044 |
| JSMA | 0.13/0.03/0.03 | **0.30**/**0.039**/0.013 | **1.00**/**0.77**/**0.50** |

## 4.3 WHITE-BOX DEFENSE

Table 3 summarizes the performance of different defenses against our white-box attack. Our white-box methods are highly effective in fooling the classifier as well as the defenses for MNIST and ASSIRA, as shown by the *before adversarial training* results. The white-box attack is unable to fool the detector for CIFAR-10 successfully.

Next, we perform adversarial training: we iteratively train the detectors against the white-box attack and allow the white-box attack access to the new defense. The white-box attack no longer successfully defeats SPD, which becomes more robust against the attack, whereas SMD is not able to become robust against the white-box attack.

Table 3: Performance of the different defenses against white-box attacks. Classifier accuracy refers to the accuracy of the image classification. Defense accuracy refers to the accuracy of the adversarial image detector (i.e. the ability to distinguish adversarial images from natural images). **The entries correspond to MNIST/CIFAR-10/ASSIRA** and the values in the parentheses below denote standard deviations.

| MODEL | DEFENSE | |
|---|---|---|
| | SMD | SPD |
| | *Before Adversarial Training* | |
| CLASSIFIER | 0.0/0.00/0.03 | 0.0/0.00/0.13 |
| | (0.00)/(0.00)/(0.01) | (0.00)/(0.00)/(0.01) |
| DETECTOR | 0.05/0.50/**0.35** | **0.18**/**0.95**/0.33 |
| | (0.02)/(0.07)/(0.06) | (0.10)/(0.01)/(0.01) |
| | *After Adversarial Training* | |
| CLASSIFIER | 0.03/0.00/0.02 | 0.00/0.00/0.04 |
| | (0.03)/(0.00)/(0.00) | (0.00)/(0.00)/(0.04) |
| DETECTOR | 0.25/0.44/0.43 | **0.79**/**0.93**/**0.71** |
| | (0.20)/(0.00)/(0.07) | (0.00)/(0.01)/(0.06) |

## 5 CONCLUSION

In our analysis, we ascertain that the saliency maps of adversarial images differ from those of natural images. Further, we show that salient pixel based defenses perform better than a saliency map defense. When trained on a single black-box attack, our method is able to detect adversarial perturbations generated by different and stronger attacks.

We show that gradients are unable to capture shifts in saliency due to adversarial perturbations and present an alternative adversarial defense using learnt saliency models that is effective against both black-box and white-box attacks. Building on the work of Gong et al. (2017), we further establish the notion of *threshold behavior*, showing that the trend depends on the $L_2$ and $L_\infty$- norms of the perturbations and therefore also prevails when using other methods (JSD) and across different attacks.

Future work could further investigate the performance of the defense in different applications. For example, as our method runs in real-time, it could be used to detect adversarial perturbations in video to counter recent attacks (Li et al., 2018; Jiang et al., 2019).

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

# A   ARCHITECTURES, HYPER-PARAMETERS AND DATA

Figure 3: ASSIRA, CIFAR-10, and MNIST image classifier architecture and hyper-parameters. The first entry corresponds to the first layer, and the table proceeds chronologically until the last layer. Parameters $f, k, p, s$ and $n$ represent the number of filters, kernel size, pooling size, stride, number of filters, respectively. If stride is omitted, it is set to 1. All classifiers have a final softmax activation.

| ASSIRA CATS AND DOGS | |
|---|---|
| LAYER | PARAMETERS |
| 2 X CONV LAYER | $f = 64, k = 3$ |
| MAXPOOLING | $p = 2, s = 2$ |
| 2 X CONV LAYER | $f = 128, k = 3$ |
| MAXPOOLING | $p = 2, s = 2$ |
| 2 X CONV LAYER | $f = 256, k = 3$ |
| MAXPOOLING | $p = 2, s = 2$ |
| 2 X CONV LAYER | $f = 512, k = 3$ |
| MAXPOOLING | $p = 2, s = 2$ |
| FLATTENING LAYER | - |
| DENSE | $n = 256$ |
| DENSE | $n = 256$ |
| DENSE | $n = 2$ |

| CIFAR-10 | |
|---|---|
| LAYER | PARAMETERS |
| 2 X CONV LAYER | $f = 32, k = 3$ |
| 2 X CONV LAYER | $f = 32, k = 3$ |
| MAXPOOLING | $p = 2, s = 2$ |
| 2 X CONV LAYER | $f = 63, k = 3$ |
| 2 X CONV LAYER | $f = 63, k = 3$ |
| MAXPOOLING | $p = 2, s = 2$ |
| 2 X CONV LAYER | $f = 128, k = 3$ |
| 2 X CONV LAYER | $f = 128, k = 3$ |
| MAXPOOLING | $p = 2, s = 2$ |
| FLATTENING LAYER | - |
| DENSE | $n = 128$ |
| DENSE | $n = 10$ |

| MNIST | |
|---|---|
| LAYER | PARAMETERS |
| 2 X CONV LAYER | $f = 64, k = 8$ |
| 2 X CONV LAYER | $f = 128, k = 6$ |
| 2 X CONV LAYER | $f = 128, k = 3$ |
| FLATTENING LAYER | - |
| DENSE | $n = 10$ |

We apply drop-out before every dense layer. Using a validation set, we experimented with different drop-out rates between $0.3$ and $0.7$ and found that the rate $\delta = 0.6$ was optimal. We use a ReLu activation for the penultimates layers and a softmax activation for the final layer. We train the model for 10 epochs on batches of size $50$.

We compare the performance of the models on MNIST, CIFAR-10 and ASSIRA (see Burges & Cortes, 1998; Krizhevsky et al., 2009; Elson et al., 2007, respectively). For MNIST and CIFAR-10, we use the standard train and test splits, and for ASSIRA, we use $3,000$ images. We use $10\%$ of the training data for the validation set, and re-train on the full training dataset once hyper-parameters were selected.

**Further experimentation of ID and JSD architecture**   We further experiment with the architectures of ID and JSD to determine whether the observed performance was the result of the architecture. In particular, we considered the adjustments as summarized in Table A; however, we found that the changes did not improve performance.

| ARCHITECTURE EXPERIMENTATION | |
|---|---|
| PARAMETER | VALUES/ VARIATION |
| DROPOUT | INCLUSION/EXCLUSION |
| OPTIMIZATION | ADAM, SGD, RMS-PROPOGRATION |
| NO. CONV LAYERS | $3 - 9$ |
| NO. HIDDEN UNITS | STARTING AT 32 OR 64, AND DOUBLING OR REMAINING CONSTANT |
| NO. DENSE LAYERS | $1 - 3$ |
| POOLING LAYERS | INCLUSION/EXCLUSION |

# B   ADVERSARIAL ATTACKS

In this section, we present the black-box adversarial attack hyper-parameters (see Figure 4), the success rates of the different adversarial attacks (see Table 4) and an example of an adversarial image generated by the various black-box attacks (see Figure 5).

Figure 4: Adversarial attack hyper-parameters. For the hyper-parameters not listed, the default values in **cleverhans** (Papernot et al., 2016a) are used. $\varepsilon$ is the maximum perturbation allowed and $\varepsilon_i$ is the maximum perturbation allowed in an iteration. We use different hyperparameters for the MNIST and CIFAR-10 to ensure the attack is sufficiently strong.

| ASSIRA | | | MNIST and CIFAR-10 | | |
|---|---|---|---|---|---|
| Attack | Norm | Hyper-parameter | Attack | Norm | Hyper-parameter |
| FGSM | $L_\infty$ | $\varepsilon = 0.025$ | FGSM | $L_\infty$ | $\varepsilon = 0.20$ |
| BIM | $L_\infty$ | $\varepsilon = 0.025, \varepsilon_i = 0.0025$ | BIM | $L_\infty$ | $\varepsilon = 0.15, \varepsilon_i = 0.015$ |
| MIM | $L_\infty$ | $\varepsilon = 0.025, \varepsilon_i = 0.0025$ | MIM | $L_\infty$ | $\varepsilon = 0.15, \varepsilon_i = 0.015$ |
| C & W | $L_2$ | binary search steps = 3 | C & W | $L_2$ | binary search steps = 3 |
| DF | $L_2$ | max. iterations = 2 | DF | $L_2$ | max. iterations = 2 |
| JSMA | N.A. | $\varepsilon = 0.025$ | JSMA | N.A. | $\varepsilon = 0.05$ |

Table 4: Accuracy of Classifier on (1) natural images (corresponding to the row *Baseline*) and adversarial images (corresponding to the remaining rows). **Entries correspond to MNIST/CIFAR-10/ASSIRA.**

| ATTACK | ACCURACY |
|---|---|
| BASELINE | 99.12/85.02/92.06 |
| FGSM | 19.27/11.11/19.21 |
| BIM | 17.07/10.16/9.21 |
| MIM | 22.00/10.09/9.21 |
| C & W | 6.86/8.99/7.94 |
| DF | 0.91/8.80/7.94 |
| JSMA | 6.05/1.86/7.94 |

| Original | FGSM | BIM | MIM | JSMA | C & W | DF |
|---|---|---|---|---|---|---|

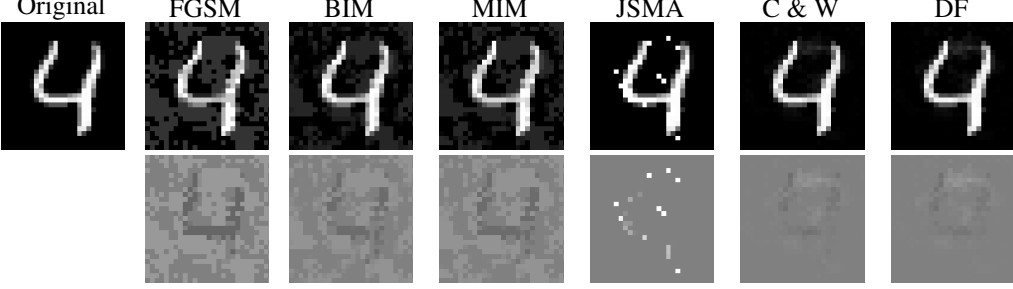

Figure 5: Example of an Adversarial Image for the MNIST dataset. From top to bottom: the top row is the set of images; the bottom row shows the size of the noise added. Gray indicates no change, whereas white indicates that the image has been made lighter, and black indicates that the image has been made darker. As MNIST images are gray-scale low-resolution images, the adversarial perturbations are perceptible to the human eye. Nevertheless, the correct classification of the image is still clearly 4. However, the classifier predicts (from left to right) the images as: 4, 9, 9 , 8, 9, 9. Further, we observe that the perturbations of FGSM, BIM, and MIM are more visible than those of C & W and DF.

## C  WHITE-BOX ATTACKS PSEUDO-CODE

---

**Algorithm 1** White-box attack for JSD

---
1: $x_{adv} \leftarrow x$
2: **for** $t = 0 : T$ **do**
3:     **for** $j = 1 : n$ **do**
4:         **if** $x_{adv}$ does not fool the classifier **then**
5:             $x_{adv} \leftarrow x_{adv} + \varepsilon\text{sign}(\nabla f(x_{adv}, y))$
6:         **end if**
7:         **if** $x_{adv}$ does not fool the detector **then**
8:             $s_{adv} \leftarrow \nabla f(x_{adv})$
9:             $[x_{adv}, s_{adv}] \leftarrow [x_{adv}, s_{adv}] + \varepsilon\text{sign}(\nabla d([x_{adv}, s_{adv}], y))$
10:            $x_{adv} \leftarrow Clip(x_{adv})$
11:         **end if**
12:     **end for**
13: **end for**
14: **return** $x_{adv}$

---

Algorithm 1 provides the white-box attack for JSD. As mentioned in Section 2, JSD concatenates the image with its saliency map (computed as the Jacobian) and uses this as an input to the classifier. Algorithms 2 and 3 provide the white-box attacks for our defenses: SPD and SMD. The function $f_s$ corresponds to generating the saliency map using the method introduced by Dabkowski & Gal (2017). Their method returns a two-dimensional saliency map. However, as the image is three dimensions, we expand the last dimension and stack the map to match the number of channels ($n_c$) of the image. In doing so, we assume that the saliency is constant along depth.

---

**Algorithm 2** White-box attack for SPD

---
1: $x_{adv} \leftarrow x$
2: **for** $j = 1 : n$ **do**
3:     **for** $t = 1 : T$ **do**
4:         **if** $x_{adv}$ does not fool the classifier **then**
5:             $x_{adv} \leftarrow x_{adv} + \varepsilon\text{sign}(\nabla f(x_{adv}, y))$
6:         **end if**
7:         $s_{adv} \leftarrow f_s(x_{adv})$
8:         $sp(x_{adv}) \leftarrow x_{adv} \cdot s_{adv}$
9:         **if** $sp(x_{adv})$ does not fool the detector **then**
10:             $r \leftarrow \varepsilon\text{sign}(\nabla d(sp(x_{adv}), y))$
11:             if $n_c > 1$, repeat $r$ along the last dimension until it matches $n_c$
12:             $x_{adv} \leftarrow Clip(x_{adv} + r)$
13:         **end if**
14:     **end for**
15: **end for**
16: **return** $x_{adv}$

---

**Algorithm 3** White-box attack for SMD

---
1: $x_{adv} \leftarrow x$
2: **for** $j = 1 : n$ **do**
3:     **for** $t = 1 : T$ **do**
4:         **if** $x_{adv}$ does not fool the classifier **then**
5:             $x_{adv} \leftarrow x_{adv} + \varepsilon\text{sign}(\nabla f(x_{adv}, y))$
6:         **end if**
7:         $s_{adv} \leftarrow f_s(x_{adv})$
8:         **if** $s_{adv}$ does not fool the detector **then**
9:             $r \leftarrow \varepsilon\text{sign}(\nabla d(s_{adv}, y))$
10:             if $n_c > 1$, repeat $r$ along the last dimension until it matches $n_c$
11:             $x_{adv} \leftarrow Clip(x_{adv} + r)$
12:         **end if**
13:     **end for**
14: **end for**
15: **return** $x_{adv}$

---

# D    SALIENCY MAPS

## D.1    MODEL-BASED SALIENCY MAPS

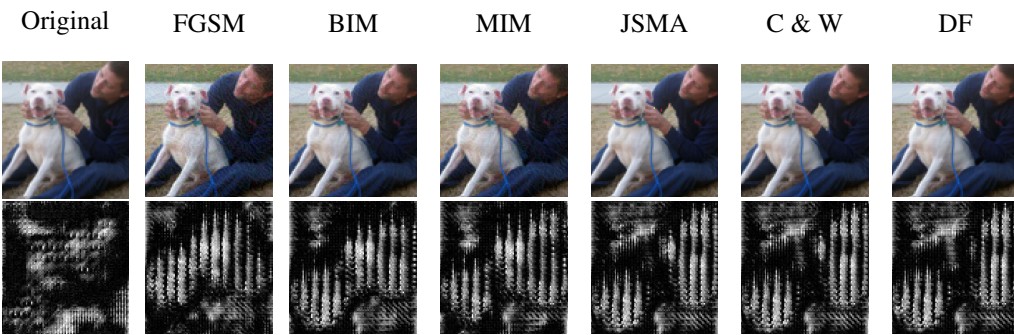

Figure 6: Example of the saliency maps of natural and adversarial images for the ASSIRA cats and dogs dataset. The top row shows the input image and the bottom shows the corresponding saliency map. In the saliency row, lighter colours correspond to higher saliency (black corresponds to a saliency of 0, the lowest possible value). The classifier predicts *dog* for the original image and *cat* for the adversarial images. Note the stark difference between the saliency masks of the original image and those of the adversarial examples.

## D.2    GRADIENT-BASED SALIENCY MAPS

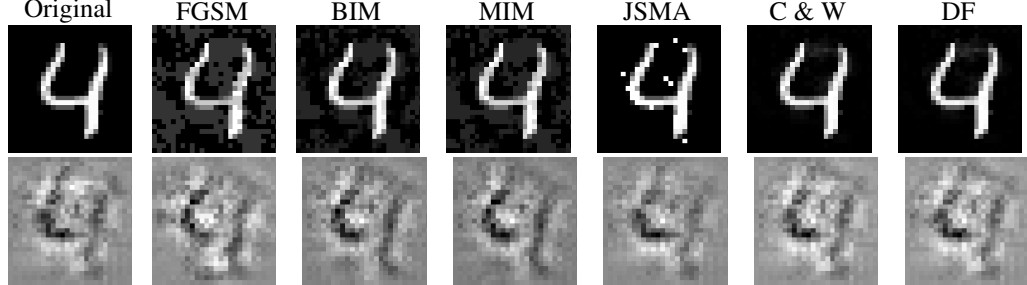

Figure 7: Examples of gradient-based saliency maps for MNIST. The first row shows the natural and adversarial images and the second row shows their respective saliency maps. Although there are slight deviations between the saliency maps, if the maps were unlabelled, it would be unclear which maps belong to adversarial examples as opposed to the original image.

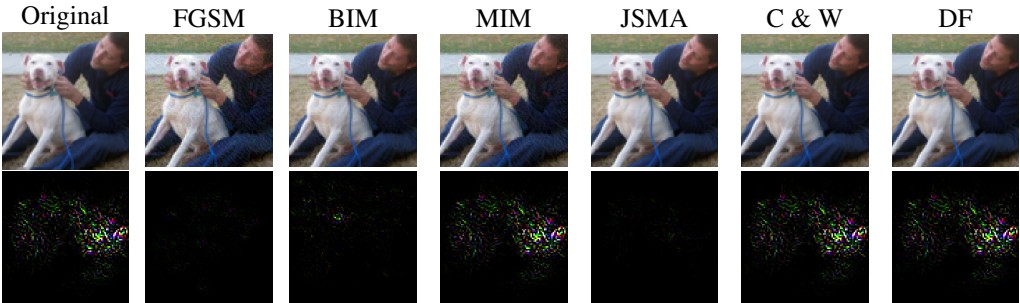

Figure 8: Examples of gradient-based saliency maps for ASSIRA. The first row shows the natural and adversarial images, and the second row shows their respective saliency maps. There are no perceptible differences between the saliency map of the original image and adversarial images generated using MIM, C&W, and DF. Further, we observe that for FGSM and JSMA, the gradients are all zero-valued. This is a second drawback of using gradients– they are unstable and generate uninformative saliency maps due to underflow.

# E $L_2$ DISTANCES BETWEEN SALIENCY MAPS CORRESPONDING TO ADVERSARIAL IMAGES GENERATED BY DIFFERENT ATTACKS

Table 5: MNIST/ASSIRA: $L_2$ distance between saliency maps generated by different adversarial attacks

|      | BIM | MIM | C &W | DF | JSMA |
|------|-----|-----|------|-----|------|
| FGSM | 0.05/0.07/0.01 | 0.04/0.07/0.01 | 0.12/0.07/0.02 | 0.07/0.08/0.02 | 0.11/0.07/0.02 |
|      | (0.03)/(0.03)/(0.01) | (0.03)/(0.03)/(0.01) | (0.04)/(0.03)/(0.01) | (0.03)/(0.03)/(0.01) | (0.01)/(0.03)/(0.03) |
| BIM  |     | 0.02/0.03/0.00 | 0.12/0.05/0.01 | 0.01/0.05/0.02 | 0.11/0.07/0.01 |
|      |     | (0.02)/(0.02)/(0.00) | (0.04)/(0.03)/(0.00) | (0.03)/(0.03)/(0.01) | (0.03)/(0.03)/(0.00) |
| MIM  |     |     | 0.12/0.06/0.01 | 0.06/0.06/0.01 | 0.11/0.07/0.01 |
|      |     |     | (0.04)/(0.03)/(0.00) | (0.03)/(0.03)/(0.01) | (0.03)/(0.03)/(0.00) |
| CW   |     |     |     | 0.11/0.02/0.00 | 0.13/0.07/0.00 |
|      |     |     |     | (0.05)/(0.03)/(0.01) | (0.03)/(0.03)/(0.00) |
| DF   |     |     |     |     | 0.10/0.02/0.01 |
|      |     |     |     |     | (0.04)/(0.03)/(0.01) |

# F SINGLE BLACK-BOX ADVERSARIAL ATTACK DETECTOR

Table 6 summarizes the accuracies of the different defenses when training a *single detector* against a combination of different types of black-box attacks. All methods perform relatively similarly as when trained against a single defense, obtaining slightly worse performances than when trained against a specific adversarial attack. This is useful in practice when it is unclear which adversarial attack is used.

Table 6: Performance of the different defenses against the combination of all black-box attacks. The entries are the accuracy of the defense in distinguishing natural and adversarial images for **MNIST/CIFAR-10/ASSIRA**. The values in parentheses below denote standard deviations.

| DEFENSE | | | |
|---------|---|---|---|
| BASELINES | | OUR METHODS | |
| ID | JSD | SMD | SPD |
| **1.00**/0.61/0.50 | 0.99/0.58/0.50 | 0.96/**1.00**/**1.00** | 0.99/**1.00**/**1.00** |
| (0.00)/(0.16)/(0.00) | (0.01)/(0.11)/(0.00) | (0.03)/(0.00)/(0.00) | (0.00)/(0.00)/(0.00) |

# G BLACK-BOX ADVERSARIAL ATTACK RESULTS

## G.1 MNIST

Table 7: The performance of the different defenses trained on a single black-box attack on detecting different black-box attacks. **The entries correspond to** the accuracy the defenses (**ID/JSD/SPD/SMD**) obtain in distinguishing adversarial examples from natural images. The values in parentheses below denote standard deviations.

| DEFENSE | FGSM | BIM | MIM | C & W | DF | JSMA |
|---|---|---|---|---|---|---|
| | | | ATTACK | | | |
| FGSM | 1.00/1.00/0.94/1.00 | 1.00/1.00/0.91/1.00 | 1.00/1.00/0.92/1.00 | 0.69/0.71/0.75/0.96 | 0.78/0.80/0.70/0.98 | 0.74/0.72/0.80/0.70 |
| | (0.00)/(0.00)/(0.00)/(0.00) | (0.00)/(0.00)/(0.02)/(0.00) | (0.00)/(0.00)/(0.02)/(0.00) | (0.03)/(0.02)/(0.01)/(0.00) | (0.05)/(0.03)/(0.03)/(0.00) | (0.02)/(0.03)/(0.00)/(0.00) |
| BIM | 1.00/1.00/1.00/0.92 | 1.00/1.00/1.00/0.91 | 1.00/1.00/1.00/0.91 | 0.74/0.75/0.97/0.77 | 0.83/0.85/.98/0.73 | 0.72/0.68/0.82/0.72 |
| | (0.00)/(0.00)/(0.00)/(0.01) | (0.00)/(0.00)/(0.00)/(0.01) | (0.00)/(0.00)/(0.00)/(0.01) | (0.06)/(0.06)/(0.01)/(0.01) | (0.06)/(0.07)/(0.01)/(0.01) | (0.02)/(0.01)/(0.02)/(0.001) |
| MIM | 1.00/1.00/1.00/1.92 | 1.00/1.00/1.00/0.92 | 1.00/1.00/1.00/0.92 | 0.72/0.72/0.96/0.76 | 0.81/0.80/0.98/0.72 | 0.72/0.70/0.73/0.81 |
| | (0.00)/(0.00)/(0.01)/(0.00) | (0.00)/(0.00)/(0.01)/(0.01) | (0.00)/(0.00)/(0.01)/(0.00) | (0.03)/(0.03)/(0.01)/(0.01) | (0.04)/(0.03)/(0.01)/(0.01) | (0.03)/(0.01)/(0.2)/(0.01) |
| DF | 1.00/1.00/0.99/0.87 | 1.00/1.00/0.98/0.87 | 1.00/1.00/1.00/0.85 | 1.00/1.00/0.99/0.95 | 1.00/1.00/0.98/0.95 | 0.74/0.71/0.83/0.92 |
| | (0.00)/(0.00)/(0.00)/(0.00) | (0.00)/(0.00)/(0.01)/(0.00) | (0.00)/(0.00)/(0.01)/(0.00) | (0.00)/(0.00)/(0.00)/(0.00) | (0.00)/(0.00)/(0.01)/(0.00) | (0.06)/(0.01)/(0.05)/(0.00) |
| CW | 1.00/1.00/0.95/0.81 | 1.00/1.00/0.94/0.83 | 1.00/1.00/0.94/0.81 | 1.00/1.00/0.94/0.94 | 1.00/1.00/0.95/0.91 | 1.00/0.74/0.84/0.87 |
| | (0.00)/(0.00)/(0.00)/(0.00) | (0.00)/(0.00)/(0.02)/(0.00) | (0.00)/(0.00)/(0.02)/(0.00) | (0.03)/(0.02)/(0.01)/(0.00) | (0.05)/(0.03)/(0.03)/(0.00) | (0.02)/(0.03)/(0.00)/(0.00) |
| SAL | 0.73/0.64/0.80/0.93 | 0.51/0.51/0.72/0.88 | 0.52/0.51/0.73/0.88 | 0.50/0.50/0.71/0.89 | 0.50/0.50/0.70/0.85 | 1.00/1.00/0.97/0.97 |
| | (0.10)/(0.05)/(0.03)/(0.01) | (0.01)/(0.01)/(0.02)/(0.02) | (0.02)/(0.01)/(0.02)/(0.01) | (0.0)/(0.00)/(0.01)/(0.00) | (0.00)/(0.00)/(0.01)/(0.01) | (0.00)/(0.00)/(0.00)/(0.00) |

## G.2 CIFAR-10

Table 8: The performance of the different defenses trained on a single black-box attack on detecting different black-box attacks. **The entries correspond to** the accuracy the defenses (**ID/JSD/SPD/SMD**) obtain in distinguishing adversarial examples from natural images. The values in parentheses below denote standard deviations.

| DEFENSE | ATTACK | | | | | |
|---|---|---|---|---|---|---|
| | FGSM | BIM | MIM | C & W | DF | JSMA |
| FGSM | 1.00/1.00/1.00/1.00 (0.00)/(0.00)/(0.00)/(0.00) | 0.50/0.50/1.00/0.99 (0.00)/(0.00)/(0.00)/(0.01) | 0.55/0.58/1.00/0.99 (0.03)/(0.04)/(0.00)/(0.01) | 0.50/0.50/1.00/0.99 (0.00)/(0.00)/(0.00)/(0.01) | 0.50/0.50/1.00/0.99 (0.00)/(0.00)/(0.00)/(0.01) | 0.67/0.60/1.00/0.99 (0.09)/(0.05)/(0.00)/(0.01) |
| BIM | 1.00/1.00/1.00/1.00 (0.00)/(0.00)/(0.00)/(0.00) | 1.00/0.99/1.00/1.00 (0.00)/(0.00)/(0.00)/(0.00) | 1.00/1.00/1.00/1.00 (0.00)/(0.00)/(0.00)/(0.00) | 0.50/0.50/1.00/1.00 (0.00)/(0.00)/(0.00)/(0.00) | 0.50/0.50/1.00/1.00 (0.00)/(0.00)/(0.00)/(0.00) | 0.90/0.71/1.00/1.00 (0.10)/(0.02)/(0.00)/(0.00) |
| MIM | 1.00/1.00/1.00/1.00 (0.00)/(0.00)/(0.00)/(0.00) | 0.62/0.67/1.00/1.00 (0.08)/(0.12)/(0.00)/(0.00) | 1.00/0.99/1.00/1.00 (0.00)/(0.00)/(0.00)/(0.00) | 0.50/0.50/1.00/1.00 (0.00)/(0.00)/(0.00)/(0.00) | 0.50/0.50/1.00/1.00 (0.00)/(0.00)/(0.00)/(0.00) | 0.82/0.81/1.00/1.00 (0.08)/(0.09)/(0.00)/(0.00) |
| DF | 0.50/0.50/1.00/1.00 (0.00)/(0.00)/(0.00)/(0.00) | 0.50/0.50/1.00/1.00 (0.00)/(0.00)/(0.00)/(0.00) | 0.50/0.50/1.00/1.00 (0.00)/(0.00)/(0.00)/(0.00) | 0.50/0.50/1.00/1.00 (0.00)/(0.00)/(0.00)/(0.00) | 0.50/0.50/1.00/1.00 (0.00)/(0.00)/(0.00)/(0.00) | 0.50/0.50/1.00/1.00 (0.00)/(0.00)/(0.00)/(0.00) |
| CW | 0.50/0.50/1.00/1.00 (0.00)/(0.00)/(0.00)/(0.00) | 0.50/0.50/1.00/1.00 (0.00)/(0.00)/(0.00)/(0.00) | 0.50/0.50/1.00/1.00 (0.00)/(0.00)/(0.00)/(0.00) | 0.50/0.50/1.00/1.00 (0.00)/(0.00)/(0.00)/(0.00) | 0.50/0.50/1.00/1.00 (0.00)/(0.00)/(0.00)/(0.00) | 0.50/0.50/1.00/1.00 (0.00)/(0.00)/(0.00)/(0.00) |
| JSMA | 1.00/0.99/1.00/1.00 (0.01)/(0.00)/(0.00)/(0.00) | 0.52/0.55/1.00/1.00 (0.03)/(0.05)/(0.00)/(0.00) | 0.80/0.89/1.00/1.00 (0.18)/(0.16)/(0.00)/(0.00) | 0.50/0.50/1.00/1.00 (0.00)/(0.00)/(0.00)/(0.00) | 0.50/0.50/1.00/1.00 (0.00)/(0.00)/(0.00)/(0.00) | 1.00/0.99/1.00/1.00 (0.00)/(0.00)/(0.00)/(0.00) |

G.3 ASSIRA

Table 9: The performance of the different defenses trained on a single black-box attack on detecting different black-box attacks. **The entries correspond to** the accuracy the defenses (**ID/JSD/SPD/SMD**) obtain in distinguishing adversarial examples from natural images. The values in parentheses below denote standard deviations.

| DEFENSE | ATTACK | | | | | |
| --- | --- | --- | --- | --- | --- | --- |
| | FGSM | BIM | MIM | C & W | DF | JSMA |
| FGSM | 0.66/0.66/1.00/1.00 (0.20)/(0.20)/(0.00)/(0.00) | 0.50/0.50/1.00/1.00 (0.00)/(0.00)/(0.00)/(0.00) | 0.50/0.50/1.00/1.00 (0.00)/(0.00)/(0.00)/(0.00) | 0.50/0.50/1.00/1.00 (0.00)/(0.00)/(0.00)/(0.00) | 0.50/0.50/1.00/1.00 (0.00)/(0.00)/(0.00)/(0.00) | 0.50/0.50/1.00/1.00 (0.00)/(0.00)/(0.00)/(0.00) |
| BIM | 0.50/0.50/1.00/1.00 (0.00)/(0.00)/(0.00)/(0.00) | 0.50/0.50/1.00/1.00 (0.00)/(0.00)/(0.00)/(0.00) | 0.50/0.50/1.00/1.00 (0.00)/(0.00)/(0.00)/(0.00) | 0.50/0.50/1.00/1.00 (0.00)/(0.00)/(0.00)/(0.00) | 0.50/0.50/1.00/1.00 (0.00)/(0.00)/(0.00)/(0.00) | 0.50/0.50/1.00/1.00 (0.00)/(0.00)/(0.00)/(0.00) |
| MIM | 0.50/0.50/1.00/1.00 (0.00)/(0.00)/(0.00)/(0.00) | 0.50/0.50/1.00/1.00 (0.00)/(0.00)/(0.00)/(0.00) | 0.50/0.50/1.00/1.00 (0.00)/(0.00)/(0.00)/(0.00) | 0.50/0.50/1.00/1.00 (0.00)/(0.00)/(0.00)/(0.00) | 0.50/0.50/1.00/1.00 (0.00)/(0.00)/(0.00)/(0.00) | 0.50/0.50/1.00/1.00 (0.00)/(0.00)/(0.00)/(0.00) |
| C & W | 0.50/0.50/1.00/1.00 (0.00)/(0.00)/(0.00)/(0.00) | 0.50/0.50/1.00/1.00 (0.00)/(0.00)/(0.00)/(0.00) | 0.50/0.50/1.00/1.00 (0.00)/(0.00)/(0.00)/(0.00) | 0.50/0.50/1.00/1.00 (0.00)/(0.00)/(0.00)/(0.00) | 0.50/0.50/1.00/1.00 (0.00)/(0.00)/(0.00)/(0.00) | 0.50/0.50/1.00/1.00 (0.00)/(0.00)/(0.00)/(0.00) |
| DF | 0.50/0.50/1.00/1.00 (0.00)/(0.00)/(0.00)/(0.00) | 0.50/0.50/1.00/1.00 (0.00)/(0.00)/(0.00)/(0.00) | 0.50/0.50/1.00/1.00 (0.00)/(0.00)/(0.00)/(0.00) | 0.50/0.50/1.00/1.00 (0.00)/(0.00)/(0.00)/(0.00) | 0.50/0.50/1.00/1.00 (0.00)/(0.00)/(0.00)/(0.00) | 0.50/0.50/1.00/1.00 (0.00)/(0.00)/(0.00)/(0.00) |
| JSMA | 0.50/0.50/1.00/1.00 (0.00)/(0.00)/(0.00)/(0.00) | 0.50/0.50/1.00/1.00 (0.00)/(0.00)/(0.00)/(0.00) | 0.50/0.50/1.00/1.00 (0.00)/(0.00)/(0.00)/(0.00) | 0.50/0.50/1.00/1.00 (0.00)/(0.00)/(0.00)/(0.00) | 0.50/0.50/1.00/1.00 (0.00)/(0.00)/(0.00)/(0.00) | 1.00/1.00/1.00/1.00 (0.00)/(0.00)/(0.00)/(0.00) |

