# OpenReview forum: "Model-based Saliency for the Detection of Adversarial Examples"
_ICLR.cc/2020/Conference — Reject_

### Official Review · AnonReviewer3 · 2019-10-18
**Official Blind Review #3**

**Rating:** 6

**Review:**

This paper presents a method for training networks to detect adversarial examples and by virtue of doing so, providing defense against adversarial attacks. Two different approaches are examined, in which a saliency map is used in combination with the input as a mask. In one instance the saliency mask is based on a classifier used to distinguish "normal" from adversarial examples. In the other instance, the salient pixels themselves form the basis for defense. In both cases, the saliency map is combined with the image for training a CNN by way of an element-wise product.
Overall, this presents a relatively simplistic way of deriving representations of saliency and combining these with inputs for training that builds robustness against white and black box attacks. At the same time, the empirical results presented reveal a considerable degree of success in providing a defense against such attacks. I find that this presents an interesting contribution to the literature addressing both adversarial attacks, and new notions on ways of characterizing saliency.

**Experience Assessment:**

I have read many papers in this area.

**Review Assessment: Checking Correctness Of Derivations And Theory:**

I assessed the sensibility of the derivations and theory.

**Review Assessment: Checking Correctness Of Experiments:**

I assessed the sensibility of the experiments.

**Review Assessment: Thoroughness In Paper Reading:**

I read the paper at least twice and used my best judgement in assessing the paper.

---

> ### Author Response · Authors · 2019-11-13
> **Response**
>
> We would like to thank the reviewer for their feedback. We would like to clarify one detail: our defense either (1) uses the saliency map or (2) uses the dot-product between the saliency map and input image. Neither includes the original image as a concatenated input.

---

### Official Review · AnonReviewer2 · 2019-10-22
**Official Blind Review #2**

**Rating:** 1

**Review:**

The paper studies methods for detecting adversarial examples using saliency maps. The authors propose using the method of Dabkowski and Gal (2017) to generate saliency maps and then train a classifier on these maps (or their dot product with the input image) to distinguish natural from adversarial examples. They perform experiments evaluating the white-box and black-box robustness of their detection scheme.

From a technical perspective, the contribution of the paper is rather incremental. The detection of adversarial examples by training a classifier on saliency maps has already been studied in prior work. The only modification proposed in this work is using an (existing) alternative method for producing the saliency maps and utilizing the dot product of maps with images.

From a conceptual perspective, the impact of detecting specific adversarial attacks is not clear. In a realistic setting, an adversary could use a very different attack or even utilize a different set of transformations (e.g. image rotations). Thus, in order to demonstrate the utility of their method in a black-box scenario, the authors would need to evaluate the defense in a variety of different scenarios. At the very least, they should consider generalization to difference attacks (e.g., train against FGSM and BIM, and test against DF).

Moreover, the robustness against white-box adversaries is not sufficiently studied. Firstly, the robustness of the non-adversarially trained detector is suspiciously high. There is little reason to expect that a composition of two neural networks (the saliency map methods and the classifier) would be non-trivially robust. The authors should consider alternative attacks perhaps using more iterations with a smaller step size. Secondly, after adversarial training, only the robustness against the same attack is considered. In order to argue about white-box robustness, the authors would need to evaluate against a variety of diverse adversaries.

Overall, the technical and conceptual contribution of this paper is insufficient for publication at ICLR, even ignoring the concerns about its experimental evaluation.

**Experience Assessment:**

I have published in this field for several years.

**Review Assessment: Checking Correctness Of Derivations And Theory:**

I carefully checked the derivations and theory.

**Review Assessment: Checking Correctness Of Experiments:**

I carefully checked the experiments.

**Review Assessment: Thoroughness In Paper Reading:**

I read the paper thoroughly.

---

> ### Author Response · Authors · 2019-11-14
> **Response**
>
> We would like to thank the reviewer for their extensive and useful feedback. With regard to the novelty of our paper, please see our general comment above. Below, we would like to address some of the further points highlighted in your review:
>
> "From a conceptual perspective, the impact of detecting specific adversarial attacks is not clear. In a realistic setting, an adversary could use a very different attack or even utilize a different set of transformations (e.g. image rotations). Thus, in order to demonstrate the utility of their method in a black-box scenario, the authors would need to evaluate the defense in a variety of different scenarios." and "Secondly, after adversarial training, only the robustness against the same attack is considered. In order to argue about white-box robustness, the authors would need to evaluate against a variety of diverse adversaries."
>
> We appreciate that ideally a defense would be suitable against all scenarios, however, in practice, it is difficult to consider all possible transformations and attacks. Recent research has focused on particular black-box and white-box attacks [e.g. Dhillon 2018; Liao 2018]. We follow the general recommendations of Carlini et al. [2017] in our evaluation protocol.
>
> "At the very least, they should consider generalization to difference attacks (e.g., train against FGSM and BIM, and test against DF)."
>
> Thank you for this feedback- we have evaluated the performance of our defense across different attacks and found that it generalizes well. We have added these results to section 4.2 (and the values can be found in Appendix G).
>
> "There is little reason to expect that a composition of two neural networks (the saliency map methods and the classifier) would be non-trivially robust."
>
> In our salient pixel technique, we compute the dot-product of the image and the saliency map. If we target the defense via a gradient-based approach, it is not directly clear how the perturbation translates to the original image. From this perspective, it can improve robustness. We would be curious to hear your further thoughts on this topic.

---

> > ### Comment · AnonReviewer2 · 2019-11-15
> > **Response**
> >
> > Thank you for the response and performing additional experimental evaluation.
> >
> > -- I agree that prior work has studied the problem of detecting and defending against specific attacks (as opposed to model general threat models such as Lp norms). However, the impact of this research is unclear, given that these defenses can be circumvented by adaptive attacks.
> > -- The fact that the defense generalized to other similar black-box attacks is encouraging. However, I still find the evidence in favor of the defense lacking. From a black-box perspective this is only one example of attack (gradient-based methods), while the white-box evaluation shows that the basic defense (without adv training) can be bypassed.
> > -- Unfortunately I do not agree with your argument. In the end, the overall defense is a complex function of the input that is likely sensitive to different perturbations. After all, the white-box evaluation performed highlights this brittleness quite clearly.
> >
> > Overall, I still do not find the evidence in favor of the method sufficient. Given that the novelty is quite limited I keep my score as is.

---

> > > ### Author Response · Authors · 2019-11-15
> > > **Response**
> > >
> > > Thanks for your quick response- we highly appreciate it.
> > >
> > > While experimenting with white-box attacks, our goal was to break our defense. As such, we generated attacks with large perturbations. (And wanted to show that iteratively, we could learn to also defend against such attacks.) However, the resultant images are unclassifiable by human standards. (please see the example uploaded at the Github link provided below . )
> > >
> > > Vis-a-vis your original comment about CIFAR-10. We have run some additional experiments, decreasing alpha (for n iterations =100). SMD performs poorly, obtaining an accuracy of 0.32.
> > >
> > > +-----+--------------------------------------+
> > > |     | Perturbation Size (alpha)            |
> > > +-----+------+-------+-------+-------+-------+
> > > |     | 0.05 | 0.005 | 0.001 | 0.005 | 0.001 |
> > > +-----+------+-------+-------+-------+-------+
> > > | ID  | 0.50 | 0.50  | 0.50  | 0.50  | 0.50  |
> > > +-----+------+-------+-------+-------+-------+
> > > | JSD | 0.50 | 0.50  | 0.50  | 0.50  | 0.50  |
> > > +-----+------+-------+-------+-------+-------+
> > > | SPD | 0.90 | 0.77  | 0.59  | 0.54  | 0.53  |
> > > +-----+------+-------+-------+-------+-------+
> > >
> > > ID = Image-based defense (Gong, 2017), JSD = Jacobian Saliency-based Defense (Zhang, 2018), SPD= Salient Pixel Defense; SMD = Saliency Map Defense.  ID/JSD deterministically predict a single class and therefore achieve 50% accuracy.
> > >
> > > Further, we've included the code for the white-box attack here: https://github.com/codeiclr2020/Model-based-Saliency-for-the-Detection-of-Adversarial-Examples. If you have any questions or would like to see the code for different experiments, please don't hesitate to ask us.

---

### Official Review · AnonReviewer1 · 2019-10-23
**Official Blind Review #1**

**Rating:** 3

**Review:**

This paper proposes an adversarial defense method that is a saliency-based adversarial example detector. The method is motivated by the well-known fact that saliency maps and adversarial perturbations are having similar mathematical formulations and derivations. By using model-based saliency maps rather than gradient-based ones, it seems to detect hard attacks with smaller perturbation size as well. As far as the authors mentioned, the proposed method is simply using different techniques to derive saliency maps compared to the previous methods.

Overall, the intuition and motivation of this paper are from the previous works and the main contribution is to use another (powerful) saliency map extractor for learning an adversarial detector. Although the overall results are improved from the previous methods, the proposed method is lack of novelty.

- For SMD (Saliency Map Defense), what is the reason that the input image is not used together? computational issue? performance degradation?
- Is it possible to train a single detector that can handle all different adversarial attacks?
- Would the distance between saliency maps from different attacks be small? How does the saliency map change under different attacks?
- Have you tried any other powerful saliency maps other than Dabkowski & Gal (2017)?

**Experience Assessment:**

I have published one or two papers in this area.

**Review Assessment: Checking Correctness Of Derivations And Theory:**

I assessed the sensibility of the derivations and theory.

**Review Assessment: Checking Correctness Of Experiments:**

I assessed the sensibility of the experiments.

**Review Assessment: Thoroughness In Paper Reading:**

I read the paper at least twice and used my best judgement in assessing the paper.

---

> ### Author Response · Authors · 2019-11-13
> **Response**
>
> We would like to thank the reviewer for their interest and feedback on our paper. With regard to the novelty of our paper, please see our general comment above. Below, we address your specific questions:
>
> "For SMD (Saliency Map Defense), what is the reason that the input image is not used together? computational issue? performance degradation? "
>
> We wanted to test whether the saliency mask itself is sufficient to detect adversarial attacks. As such, its performance can be seen as a ``sanity check" to verify that our saliency technique (adapted from Dabkowski & Gal [2017]) is able to capture shifts in attention due to adversarial perturbations.
>
> "Is it possible to train a single detector that can handle all different adversarial attacks?"
>
> On the attacks we evaluated, yes. Appendix E shows the results of a single classifier trained on a different type of attacks.  Further (these are new results from experiments run based on your feedback),  in Appendix G (and presented in Section 4.2), we show that SPD is successful at detecting CW and DF, even when trained on weaker attacks.
>
> "Would the distance between saliency maps from different attacks be small? How does the saliency map change under different attacks?"
>
>   Thanks for this interesting question -- we've added results to Appendix E, which provide these distances. The saliency maps do vary across different attacks. In general, they seem to be similar to the distance between the saliency map of an adversarial image and natural image.
>
> "Have you tried any other powerful saliency maps other than Dabkowski & Gal (2017)?"
>
> Our main goal was to show that (1) using salient pixels as a defense is effective and (2) one should still exhibit caution when using other saliency techniques -- a straightforward approach using gradients is unable to capture perturbations (this was shown qualitatively in parallel work by Gu & Tresp [2019], but we verify it via experiments). Other techniques are also able to capture the shifts. The two main candidates are those highlighted by Fong and Vedaldi [2017] and Gu & Tresp [2019]. We did not choose the former due to the high computational costs required to compute the kernel for every pixel at every step. The latter work was completed in parallel (published on ArXiv in late August) and to the best of our knowledge, does not yet have an online version.

---

### Author Response · Authors · 2019-11-13
**Changes to Paper and Contributions**

We would like to thank the reviewers for their feedback. As requested by reviewers 1 and 2, we have run additional experiments, where we assess the performance of the model trained on a specific adversarial attack to see how well it detects other adversarial attacks. The additional results can be found in Section 4.2 on page 7 (and in Appendix G).

Further, we have reworded the abstract to better highlight our contributions, and would like to clarify how our work fits into the literature and the novel aspects of our work:

- We show that the previously suggested realtime saliency-based defense (Zhang 2018) is not effective -- it does not improve on a simple baseline which uses the same model but with the saliency map removed. We show that this can be attributed to the weak gradient-based saliency technique used in (Zhang 2018), and propose an effective fix using a realtime model-based saliency technique.

- We introduce a new realtime defense: the salient pixel defense, which is able to successfully detect a large range of adversarial attacks. This defense performs better than using the saliency maps themselves even when model-based saliency is used.

- Fong and Vedaldi (2017) show that their saliency method is able to detect adversarial perturbations. However, the method is not suited for practice, due to the high computational costs associated with computing the kernel at every iteration for every pixel. Further, we provide a systematic analysis against a variety of adversarial attacks, including more difficult attacks such as Carlini & Wagner attack, as opposed to only BIM (as suggested by Carlini et al. [2017]). As shown by Carlini et al. [2017], most defenses break against the C & W attack whereas our method is effective.

- Parallel work to ours [Gu & Tresp, 2019] showed that their technique can capture a shift in saliency due to adversarial perturbations qualitatively, but do not provide any quantitative experiments.

---

### Decision · Program_Chairs · 2019-12-19

**Decision:**

Reject

**Comment:**

This submission proposes a method for detecting adversarial attacks using saliency maps.

Strengths:
-The experimental results are encouraging.

Weaknesses:
-The novelty is minor.
-Experimental validation of some claims (e.g. robustness to white-box attacks) is lacking.

These weaknesses were not sufficiently addressed in the discussion phase. AC agrees with the majority recommendation to reject.